# A Self-Supervised Framework for Human Instance Segmentation

**Abstract.** Existing approaches for human-centered tasks such as human instance segmentation are focused on improving the architectures of models, leveraging weak supervision or transforming supervision among related tasks. Nonetheless, the structures are highly specific and the weak supervision is limited by available priors or number of related tasks. In this paper, we present a novel self-supervised framework for human instance segmentation. The framework includes one module which iteratively conducts mutual refinement between segmentation and optical flow estimation, and the other module which iteratively refines pose estimations by exploring the prior knowledge about the consistency in human graph structures from consecutive frames. The results of the proposed framework are employed for fine-tuning segmentation networks in a feedback fashion. Experimental results on the OCHuman and COCOPersons datasets demonstrate that the self-supervised framework achieves current state-of-the-art performance against existing models on the challenging datasets without requiring additional labels. Unlabelled video data is utilized together with prior knowledge to significantly improve performance and reduce the reliance on annotations.

**Keywords:** Instance Segmentation, Prior Knowledge, Self-supervised

## 1 Introduction

In recent years, the computer vision community has devoted great efforts in acquiring understandings of human from images. Typical applications include human instance segmentation which predicts human masks [14] [37], pose estimation which detects body joints as keypoints [16], [34], [2], [10], [32], [28], [23], and human parsing which performs pixel-level analysis [8], [20], [22], [15], [36]. The three lines of research play a crucial role in intelligent surveillance systems. This study concentrates on human instance segmentation and leverages prior knowledge to reduce the need for annotations while improving generalization.

Existing research have explored either new model structures [5] or feature propagation methods [13]. However, the generalization capability of models cannot be greatly improved due to the domain discrepancy between training data and real-world test data. For intance, the identities in input images are with a limited set of poses, a model cannot detect a human with the pose that does not appear in training data. Moreover, model architectures are also limited by the

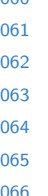
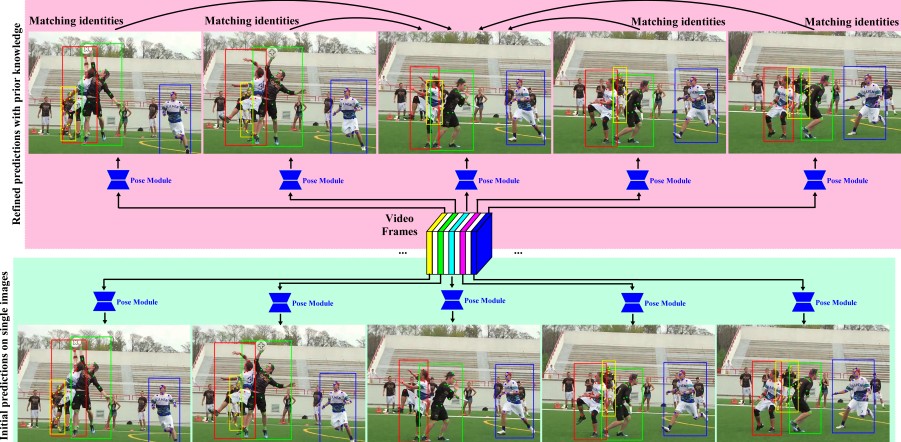

**Fig. 1.** The rationale behind the proposed approach. All the images are without any annotations. The training data of Pose Module covers limited poses and does not include the special cases in test images. For instance, the person which is occluded by another one in red bounding box in the middle image from the bottom row cannot be detected because this scenario with severe partial occlusion is not included in training data. However, if we refer to the predictions from neighboring frames, the occluded person can be recovered. The scenarios in adjacent frames are more similar to those in training data and the same person can be detected in those frames. The trajectories estimated from consecutive frames facilitate the recovery. In this way, the mistakes in the bottom middle image from test set can be fixed by the prior knowledge about motion consistency in videos. By fine-tuning on the recovered predictions, the Pose Module can generalze to the cases in test set which are not included in training data.

available training data, a typical example is the NAS-based model Auto-Deeplab [25] which was built by searching over the network space and maximizing the accuracy on training data. Nonetheless, the optimal architecture on training data leads to suboptimal performance on test set. Even if some weakly supervised methods [13] augmented supervision by exploiting the relations between different tasks, they suffer from the upper limit on the number of related tasks. The performance cannot be further improved because the approach for training with unlabelled data is under-explored.

Existing solutions to improve generalization include employing more generalizable backbones [3] [17] which were pre-trained on larger classification datasets, revising loss functions and resorting to prior knowledge [15] [20], [9]. However, the above-mentioned methods cannot resolve the challenging cases and the priors can only function as weak constraints. To significantly improve generalization by leveraging prior knowledge, we present a novel self-supervised framework for instance-segmentation. The framework is able to be trained on real-world unlabelled video sequences and achieves improvement on test set. The rationale behind the proposed framework is shown in Fig. 1.

The human-centered images can be regarded as vetors in a high-dimensional manifold. Different from common distances such as Wasserstein metrics which consider the intensities on all pixels, we measure the distances between images by

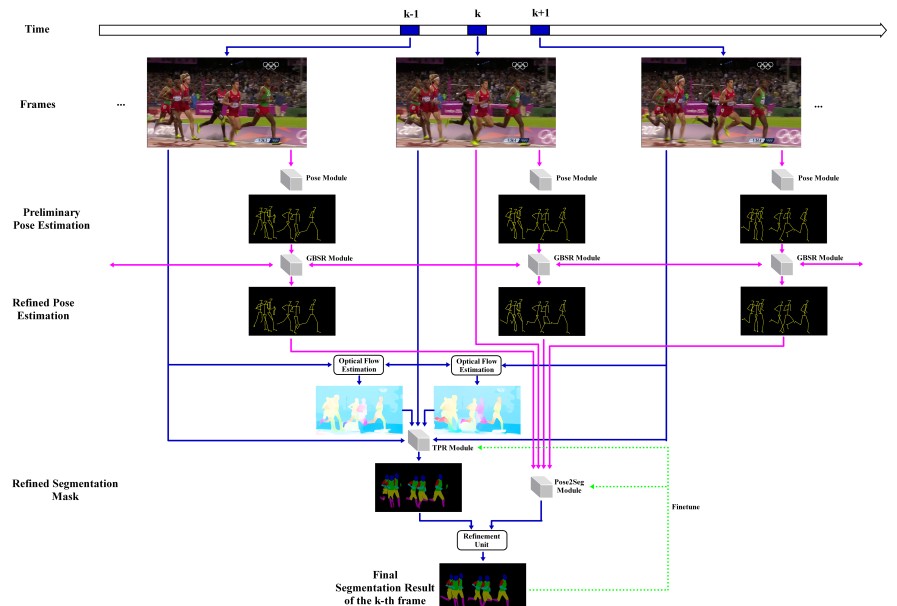

**Fig. 2.** The proposed framework for human instance segmentation. It is composed of a temporal parsing refinement module (TPR Module) for mutually refining segmentation masks and optical flow estimations, a graph-based skeleton refinement module (GBSR Module) for iteratively conducting graph distances minimization and poes estimation. The refined pose estimations are then converted to parsing results using pose-to-segmentation module (Pose2Seg Module) whose outputs are combined with those from TPR Module to produce final predictions. The final results are leveraged to fine-tune segmentation networks. The full details of GBSR Module and TPR Module are demonstrated in Fig. 3 and Fig. 5 (a), respectively.

using the similarities in human structures (poses) and human appearances. As is shown in the bottom row in Fig. 1, the model cannot detect all identities in the middle image because the patterns of poses did not appear in training images. However, the missed identity can be detected in adjacent frames. By recovering the missed detections with the consistency in temporal movements and fine-tuning the Pose Module with the recovered predictions, the Pose Module can generalize from familiar cases to previously unfamiliar cases.

As is shown in Fig. 2, the proposed framework is composed of a GBSR Module, a TPR Module and a Pose2Seg Module. The unlabelled videos are collected online resources which include [1] and other online videos. In the self-supervised training phase, the Seg Module in TPR Module and Pose Module in GBSR Module firstly conduct inference on each frame. Optical flow estimation is also conducted [18] in this phase. Then segmentation refinement and optical flow refinement are alternately conducted and mutually benefit each other. In this way, the TPR Module iteratively tackles two coherent goals: minimizing cross-entropy loss [3] for segmentation and minimizing matching error for optical flow estimations, as will be introduced in section 3.3.

In pose estimation, the GBSR Module builds a graph for each detected human, the attributes of each node in a graph include both the appearance of the semantic part and its connections with other nodes. The distances between corresponding graphs in adjacent frames are minimized with the aim of refining pose estimations. Furthermore, a pose-to-segmentation module (Pose2Seg Module) is proposed to convert the corrected skeletons to segmentation masks and the generated masks are merged with the output of TPR Module to generate the final corrected prediction. The final predictions are utilized to fine-tune the weights in Seg Module and Pose2Seg Module under a feedback fashion. The overall process is conducted for several rounds until the outputs of the Seg Module approximates the final predictions.

Our major contributions can be summarized as follows:

1. We propose a novel self-supervised framework which can be trained on unlabelled video data iteratively and improves the performance of instance segmentation with the prior knowledge about videos.
2. We propose a TPR Module which conducts mutual refinement between segmentation and optical flow estimation. Different from other methods which leverage optical flow estimations without remedying errors, the two tasks in TPR Module benefit from each other and the TPR Module facilitates the propagation of predictions from simple frames to challenging frames in the same video, as will be shown in Section 3.3.
3. We propose a novel graph based module, called GBSR Module, which tackles the goals of finetuning the pose estimation network and graph distance minimization alternately and boosts the performance in pose estimation.
4. We demonstrate the effectiveness of the framework, it achieves current state-of-the-art performance without requiring additional labels.

## 2   Related Work

**Instance Segmentation** In this task, a single mask is asigned for each object in an image. Existing deep learning methods for instance segmentation are divided into two categories. The first type of methods are composed of more than one stage. Detection is conducted before segmentation [7] [11] [35] [16]. The second type of models jointly conduct detection and segmentation in one pass [19] [26]. For instance, [26] grouped the detected line segments into connected components before figuring out object boundaries. [14] unified semantic segmentation and instance-aware edge detection in an end-to-end pass. A typical shortcome of the two-stage methods lies in their failure in detection when nearby bounding boxes are highly overlapped. Besides, different stages are trained using independent targets and their predictions are inconsistent. Even if the second type of methods do not rely on bounding boxes, some of them are composed of several sub-networks [14]. As a result, the great number of learnable parameters easily leads to over-fitting. The OCHuman and COCOPersons datasets [24] are introduced by [35]. In this paper, we introduce a self-supervised framework for instance segmentation and human parsing, unlabelled images from real-world scenarios

contribute to generalization. Besides, the proposed framework is not built on detection modules.

**Human Pose Estimation** The large datasets such as COCO Key-points Challenge have contributed to the remarkable progress in human pose estimation [30] [3] [4] [15] [21] [22] [27] [31] [37] [33]. Existing approaches can be divided into top-down [30] and bottom-up methods [28]. The former localize bounding boxes before estimating the poses inside boxes. However, challenging cases with occlusion, complex lightening conditions or entanglement usually lead to the failures of detectors. The missed detections cannot be recovered by pose estimating models. Even when parts of occluded humans are detected, the accuracy of predictions is unsatisfactory and the precision of pose estimation also drops significantly. The efficiency of top-down approaches is also inferior to bottom-up methods because their inference time is proportional to the number of people in images. Bottom-up methods predict the locations of body joints before organizing them into human structures. The computational burdens of bottom-up methods are not influenced by the number of identities in an image. However, the limbs belonging to different humans are easy to be mixed because adjacent identities are highly entangled. Additionally, the variations in scales and poses lead to the failures in organizing joints into people. To improve the robustness to occlusions and entanglement while improving generalization, we propose a GBSR Module which leverages the consistency between human poses in adjacent frames as a constraint, and refines pose predictions to meet the constraint.

## 3 Self-supervised Framework

### 3.1 The structure of the framework

The framework is composed of a GBSR Module, a TPR Module and a Pose2Seg Module. The GBSR Module and the TPR Module are introduced in Section 3.2 and Section 3.3, respectively. The method for integrating the modules are introduced in Section 3.4. The deeplabv3+ model [5] is incorporated in TPR Module as the Seg Module while the model proposed in [28] is incorporated in GBSR Module as the Pose Module. The Pose2Seg Module has the same structure as the Seg Module.

The available training data for pose estimation is significantly larger than that for segmentation. As a result, the Pose Module has a better generalization capacity and thus its predictions are leveraged to improve the segmentation performance after the post-processing of Pose2Seg Module.

### 3.2 Graph-Based Skeleton Refinement Module (GBSR Module)

The GBSR Module is introduced in Fig. 3. In the inference on single RGB images, some body joints cannot be detected due to occlusions, motion blurs or complex lightening conditions. However, the influences change from frame to frame and the same keypoint is unlikely to be missing in consecutive frames. The GBSR

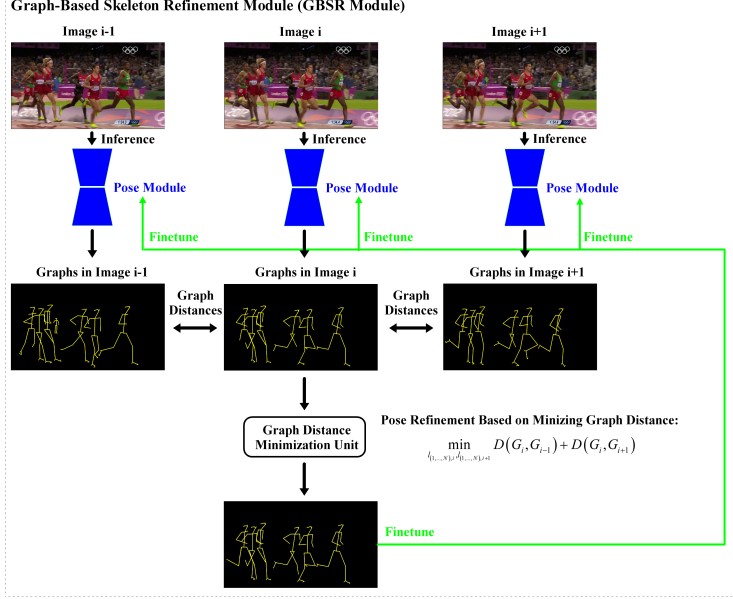

**Fig. 3.** The structure of the proposed GBSR Module.

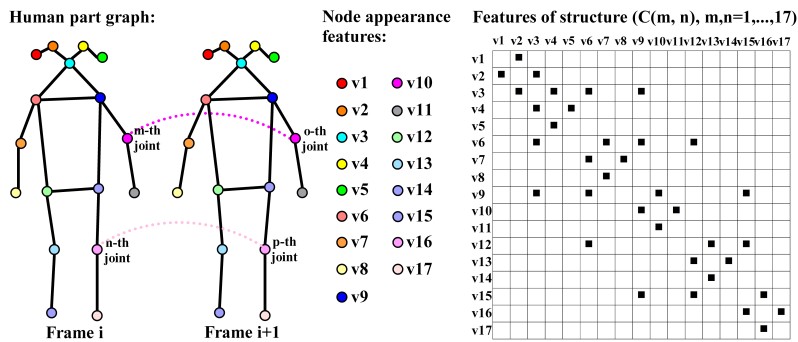

**Fig. 4.** The structure of a human graph where the representations of both appearances and structures are involved.

Module builds one graph for each human and enforces the consistency between corresponding graphs in consecutive frames by minimizing graph distances. The minimization serves the purpose of refining pose estimations. Each node in a human graph corresponds to one keypoint (body joint) and the attributes of a node involve both the appearances of the joint and its structural information such as the connections between the node and other nodes. Fig. 3 demonstrates the workflow of the GBSR Module and Fig. 4 shows human part graphs, the cells in the right table are marked in black if correponding nodes are connected.

Suppose that $G_i$ and $G_{i+1}$ are the two graphs describing the same person in the $i-th$ and $(i+1)-th$ frames. Each graph is composed of $N = 17$ nodes if without occlusion. The 17 nodes are nose, left and right eyes, left and right

ears, left and right shoulders, left and right elbows, left and right wrists, left and right hips, left and right knees, left and right ankles. The distance between two graphs is the sum of two parts

$$D(G_i, G_{i+1}) = (1 - \alpha) * L_1(G_i, G_{i+1}) + \alpha * L_2(C_i, C_{i+1}) \tag{1}$$

The first term $L_1(G_i, G_{i+1})$ measures the similarity in appearances:

$$L_1(G_i, G_{i+1}) = \sum_{m,o} v_{m,o} d(f_i(l_{m,i}), f_{i+1}(l_{o,i+1})) \tag{2}$$

where $l_{n,i}$ and $l_{n,i+1}$ denote the predicted locations of the $m-th$ and $o-th$ body joints in the $i-th$ and $(i+1)-th$ frames, respectively. A feature extraction stage consisting of 14 convolutional layers is employed to obtain the featuremaps of both frames $f_i$ and $f_{i+1}$. $f_i(l_{n,i})$ and $f_{i+1}(l_{n,i+1})$ are obtained by cropping a bounding box with an appropriate side length (twice the distance between neck and nose) from the predicted locations on feature maps and input images. $f_i(l_{n,i})$ and $f_{i+1}(l_{n,i+1})$ include both low-level and high-level contextual cues. The second term $L_2(C_i, C_{i+1})$ measures the similarity in graph structures. The structure of each graph is described by a matrix which is shown by the right column in Fig. 4. $C_i$ and $C_{i+1}$ are two $N-by-N$ matrices and $C_i(m, n) = 1$ if there is connection between the $m-th$ and the $n-th$ body joints, $m, n, o, p = 1, ..., N$, the arrangements of indices are shown in the left part of Fig. 4. $L_2(C_i, C_{i+1})$ is computed by

$$L_2(C_i, C_{i+1}) = \sum_{m,o} v_{m,o} \sum_{n,p} v_{n,p} d(C_i(m, n) - C_{i+1}(o, p)) \tag{3}$$

$d()$ is implemented using $1-$ norm, $v_{m,n}$ and $v_{o,p}$ denote the visibility scores of different body joints and ranges from 0 to 1. For instance, $m$ and $n$ denote the joints with same semantic meaning in two frames, $v_{m,n}$ is higher only when both of them are visible. As people in consecutive frames have quite similar poses, the Graph Distance Minimization Unit minimizes the distances with respect to visibility scores:

$$\min_{v_{m,o}, v_{n,p}, m, n, o, p \in 1, ..., N} D(G_i, G_{i-1}) + D(G_i, G_{i+1}) \tag{4}$$

Visibility scores are obtained in this way and are used to adjust the side lengths of boxes for cropping regions around body joints. For instance, the size of a box descreases if its visibility score is lower. The regions cropped with new sizes are leveraged in the matching of joints for a second time. Then the matching results are used to refine body joint predictions.

### 3.3   Temporal Parsing Refinement Module (TPR Module)

The details of the TPR Module is demonstrated in Fig. 5. Fig. 5 (a) shows the structure of the TPR Module, it includes a segmentation module, a unit for optical flow estimation, a unit for optical flow refinement and a unit for segmentation mask refinement. The temporal window size shows the number of

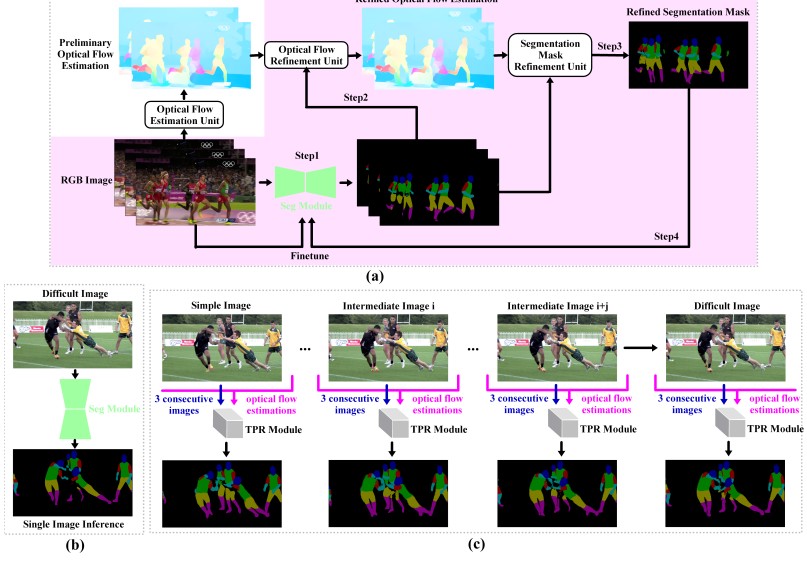

**Fig. 5.** Introduction to the proposed TPR Module. (a) Structure of the TPR Module. The shaded area denotes the iteration steps. (b) Applying the Seg Module on a single challenging frame. (c) The application of TPR Module in a video sequence where the last frame is the same as the input of (b), the consistency among consecutive frames is improved by refinement and the performance on that frame is significantly improved.

consecutive frames which compose the input to TPR Module, it is selected to be three for clear demonstration.

Firstly, optical flow estimation is conducted and the segmentation module predicts three consecutive masks. The optical flow vectors which are inconsistent with the predicted masks are regarded as unreliable vectors and are fed into the Optical Flow Refinement Unit which re-conducts a search over surrounding regions and minimizes pixel-level matching error. For the first frame and the last frame, the activations before softmax layer are warped to the middle frame based on refined flow vectors, then the element-wise sum of activations from three consecutive frames are fed to the Softmax layer (5) to produce the output. Finally the refined segmentation mask is leveraged to finetune the segmentation module. The 4 steps in the shaded area in Fig 5 (a) are conducted iteratively. The size of temporal window can be revised to adjust the dependency among consecutive predictions, as will be shown in experiments.

$$Softmax_i = Softmax(a_t(x_i, y_i) + a_{t+1}(x_i + u_{t+1}(x_i, y_i), y_i + v_{t+1}(x_i, y_i)) +$$
$$a_{t-1}(x_i - u_t(x_i, y_i), y_i - v_t(x_i, y_i))) \quad (5)$$

where $Softmax_i$ denotes the $i-th$ pixel on the output of softmax layer, $a_t$ and $a_{t+1}$ denote the activations before softmax layers in the $t-th$ and $(t+1)-th$ frames. The corrected optical flow vectors on the $i-th$ pixel are $(u_t(x_i, y_i), v_t(x_i, y_i))$

and $(u_{t+1}(x_i, y_i), v_{t+1}(x_i, y_i))$. The rationality behind applying iterations in Fig. 5 (a) is the fact that optical flow estimations are quite noisy and many estimated motion vectors are incorrect. The input in Fig. 5 (b) is the same as the last input in Fig. 5 (c). The improvements demonstrate the merits of mutual refinement which improves the consistency among consecutive predictions.

Suppose that image A is easy to conduct segmentation on while image B is challenging. A and B belong to the same video sequence. The refined optical flow estimations in Fig. 5 (a) implicitly facilitate the propagation of predictions from A to B and obtain better results on B. The propagation can be expressed in the following form

$$I_A(x, y) = I_B(x + u(x, y), y + v(x, y)), x \in [1, H], y \in [1, W] \tag{6}$$

where $I_A$ and $I_B$ denote the segmentation masks of RGB images A and B with width $W$ and height $H$. $I_A(x, y)$ shows the color intensity at location $(x, y)$ in image A. The pixel $(x, y)$ belongs to a certain semantic part and the pixel moves to a different location $(x + u(x, y), y + v(x, y))$ in image B. The transformation of human poses from A to B is divided into many intermediate steps each of which corresponds to the refined optical flow estimations in one frame. $u(x, y)$ and $v(x, y)$ are achieved by integrating the refined vectors from all intermediate steps:

$$u(x, y) = \sum_{t=1}^{T} u_t(x, y) \tag{7}$$

$$v(x, y) = \sum_{t=1}^{T} v_t(x, y) \tag{8}$$

A series of intermediate refined segmentation masks between $A$ and $B$ are obtained, such as $I_{A+1}(x, y)$, ..., $I_{A+T-1}(x, y)$ which satisfy

$$I_{A+t}(x, y) = I_{A+t+1}(x + u_{t+1}(x, y), y + v_{t+1}(x, y)), t = 1, ..., T - 1 \tag{9}$$

where the motion vectors $u_{t+1}(x, y)$ and $v_{t+1}(x, y)$ are the refined vectors in the $(t+1) - th$ step. The refined mask $I_B$ contributes to improvements in Fig. 5 (c).

## 3.4   The method for combining TPR Module with GBSR Module

Even if the TPR Module proposed in section 3.3 contributes to improving segmentation performance. There are still some limbs which cannot be detected. On the other hand, The available data for training GBSR Module [24] is different from that for training TPR Module [14] and both modules are better at handling different cases. As a result, it is necessary to combine the predictions from GBSR Module and those from TPR Module due to their complementary nature.

A pose-to-seg module (Pose2seg Module) is trained in this phase, it has the same structure as the Seg Module except for the input layer which takes in the

---

**Algorithm 1** The pipeline of the proposed approach.

---

  **Input:** The number of rounds $N_{TPR} = 5$ for mutual refinement in TPR Module, the number of rounds $N_{GBSR} = 1$ in GBSR Module for alternate between finetuning Pose Module and conducting graph distance minimization (pose refinement). The Temporal Window Size $Win_{temporal} = 3$ in both modules.

  **Output:**  Predictions on the test sets of benchmark data.

1: Select $Win_{temporal} = 3$ consecutive frames as one group. Apply the Seg Module to generate initial segmentation predictions. Also apply the Optical Flow Estimation Unit to obtain initial optical flow estimations.

2: Obtain refined segmentation masks by alternately conducting optical flow refinement with segmentation predictions and fine-tuning Seg Module using refined segmentation masks.

3: alternately conduct minimization on graph distances by refining pose estimations and re-training Pose Module.

4: Apply Pose2seg Module to generate segmentation masks using the output from Step 3. Combine the outputs from TPR Module and Pose2Seg Module using (10).

5: If the output of Step 4 and that of the Pose Module are similar enough (intersection over union above 0.95), apply the current set of learnable parameters in Seg Module to make predictions on test set and go to Step 6. Else go back to Step 1.

6: **return** Predictions on test set.

---

concatenation of RGB images and skeleton predictions. The activations before the softmax layer in Pose2Seg Module and those before the softmax layer in TPR Module are combined according to (10) to obtain the final predictions.

$$Softmax_i = Softmax(a_{TPR}(x_i, y_i) + a_{Pose2Seg}(x_i, y_i)) \qquad (10)$$

The final predictions obtained by (10) are used to re-train the learnable parameters in the TPR Module and those in the Pose2Seg Module. the pipeline of the proposed approach is presented in Algorithm 1.

## 4   Experiments

The proposed framework is evaluated on two tasks: instance segmentation and human parsing.

In instance segmentation, the framework is evaluated on two datasets: (1) COCOPersons is a subset of the MSCOCO dataset [24] and contains 64,115 images with 273,469 labelled humans. (2) OCHuman dataset which is proposed in [35] includes 4,731 images with 8,110 labelled humans. The humans in the dataset are heavily occluded and the dataset is challenging. The two datasets are among the largest ones with annotations on both human instance segmentation and pose estimation. The criterion for evaluating segmentation performance is Average Precision (AP). The MSCOCO dataset is split into three subsets: images with small objects, images with medium objects and those with large objects. The corresponding metrics are $AP_S, AP_M$ and $AP_L$. The OCHuman dataset is

divided into two subsets: OCHuman-Moderate and OCHuman-Hard, the first subset contains instances with MaxIoU in the range of 0.5 and 0.75 while the second contains instances with MaxIoU larger than 0.75, the second is more challenging. The metrics are $AP_M$ and $AP_H$, respectively.

The online collected videos include a comprehensive set of poses and actions, such as sport events, daily exercises and so on. The annotations are automatically generated with the proposed modules. Over 1,100 high-quality videos are collected in this way and will be released after publication.

### 4.1  Implementation Details

The Seg Module in TPR Module is with Deeplab-V3+ [5] structure and xception-71 [6] backbone. Firstly the Seg Module is trained on the benchmark datasets for 30 epoches with initial learning rate 1e-2 and a polynomial learning rate policy. In each round of mutual refinement, fine-tuning is conducted for 10 epoches. The Pose Module is directly inherited from [28] and it is trained on the COCOPersons dataset [24]. The initial learning rate is 2e-4. The learning rate is decayed by 0.1 after 33 epoches and ends after 40 epoches for instance segmentation.

Our performance of instance segmentation is compared with that of the Mask-RCNN model [16] and the model proposed in [35]. The Mask-RCNN model is trained with the configurations provided by the official website [12]. Resnet-50 [17] is the backbone of Mask-RCNN and the initial learning rate is 2e-2. We have also re-implemented the model proposed in [35] with official settings. The refinements are conducted until convergence, according to Algorithm 1. The training data in each round consists of the combination of the benchmark datasets and the refined segmentation results on the video dataset.

### 4.2  Performance comparison on the heavily occluded human data for instance segmentation

In this experiment, we compare the performance of the proposed framework with that of existing methods on the OCHuman dataset with occlusion cases. In this section we fix the number of rounds in mutual refinement and the temporal window size in TPR Module to be both 3, the number of refinements in GBSR Module is fixed to be 1. More choices will be discussed in section 4.5. From Table 1 it can be seen that the performance measured in $AP$ is improved by over 4 percent over existing methods on the validation and test set. Some subjective results are shown in Fig. 6.

Besides segmentation, our proposed GBSR Module can also improve the performance of keypoint localization over the Pose Module. The Pose Module achieves 0.285 / 0.303 AP on the val/test set of the OCHuman dataset, the refinement introduced in GBSR Module improves the performance to 0.299 and 0.318, respectively.

Additionally, the $\alpha$ in (1) is selected to be 0.5 because other values including 0.1, 0.2, 0.3, 0.4, 0.6, 0.7, 0.8 and 0.9 all produce inferior $AP$ than 0.5.

**Table 1.** Performance comparison on the validation and test set of the OCHuman dataset [35].

| Methods | Backbone | $AP$ val | $AP_M$ val | $AP_H$ val | Methods | Backbone | $AP$ test | $AP_M$ test | $AP_H$ test |
|---------|----------|------|------|------|---------|----------|------|------|------|
| Mask-RCNN[16] | Resnet50 | 0.163 | 0.194 | 0.113 | Mask-RCNN[16] | Resnet50 | 0.169 | 0.189 | 0.128 |
| Pose to Seg[35] | Resnet50 | 0.222 | 0.261 | 0.150 | Pose to Seg[35] | Resnet50 | 0.238 | 0.266 | 0.175 |
| Ours | Resnet50 | 0.267 | 0.310 | 0.181 | Ours | Resnet50 | 0.272 | 0.305 | 0.194 |

### 4.3  Performance comparison on general human data for instance segmentation

The COCOPersons dataset [24] is the existing largest dataset for human instance segmentation and includes all types of scenarios. The comparison is conducted on the whole dataset, the subset with medium objects and the subset with large objects. Table 2 demonstrates the results. Training is conducted on the training split and the model is evaluated on the validation set.

### 4.4  Ablation study

**4.4.1  The number of rounds for fine-tuning** As is introduced in section 4.2, the pipelines of TPR Module and GBSR Module consist of iterations. In this section, we evaluate the influence of the number of iterations on performance. Table 3 shows the results on the OCHuman dataset and Table 4 shows the results on the COCOPersons dataset.

From Table 3 and Table 4 it can be inferred that the increase in the round number contributes to improving performance. Besides, the advantage of more

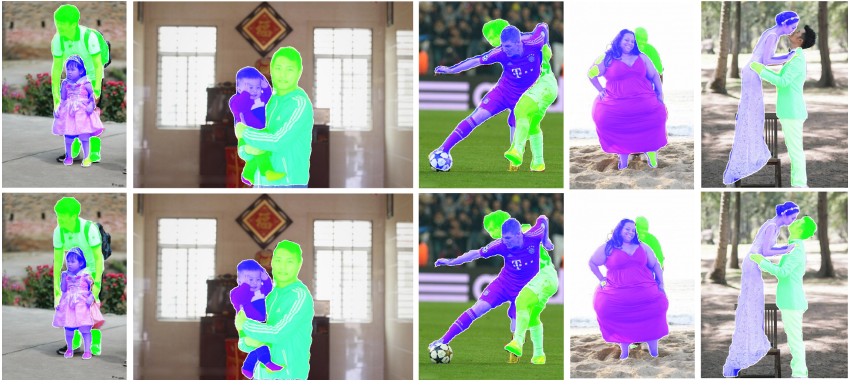

**Fig. 6.** Subject results on the OCHuman dataset [35] with occluded humans. On the top row are the predictions from the current state-of-the-art model [35]. The results of the proposed framework are shown in the bottom row. It can be seen that occlusions are better handled and background interferences are eliminated.

**Table 2.** Performance comparison on the COCOPersons dataset [35].

| Methods | Backbone | $AP$ | $AP_M$ | $AP_L$ |
|---------|----------|------|--------|--------|
| Mask-RCNN[16] | Resnet50 | 0.532 | 0.433 | 0.648 |
| PersonLab[29] | Resnet101 | - | 0.476 | 0.592 |
| PersonLab[29] | Resnet101 | - | 0.492 | 0.621 |
| PersonLab[29] | Resnet152 | - | 0.483 | 0.595 |
| PersonLab[29] | Resnet152 | - | 0.497 | 0.621 |
| Pose to Seg[35] | Resnet50 | 0.555 | 0.498 | 0.670 |
| Ours | Resnet50 | 0.626 | 0.565 | 0.714 |

**Table 3.** Performance comparison on the validation / test set of the OCHuman dataset [35] over the numbers of iterations $N_{TPR}$ and $N_{GBSR}$ in TPR Module and GBSR Module.

| $N_{TPR}$ | $N_{GBSR}$ | **Split** | $AP$ | $AP_M$ | $AP_H$ | $N_{TPR}$ | $N_{GBSR}$ | **Split** | $AP$ | $AP_M$ | $AP_H$ |
|-----------|------------|-----------|------|--------|--------|-----------|------------|-----------|------|--------|--------|
| 1 | 1 | val | 0.250 | 0.291 | 0.169 | 1 | 2 | val | 0.250 | 0.291 | 0.169 |
| 3 | 1 | val | 0.262 | 0.305 | 0.177 | 3 | 2 | val | 0.262 | 0.305 | 0.177 |
| 5 | 1 | val | 0.267 | 0.310 | 0.181 | 5 | 2 | val | 0.267 | 0.310 | 0.181 |
| 1 | 1 | test | 0.260 | 0.291 | 0.187 | 1 | 2 | test | 0.260 | 0.291 | 0.187 |
| 3 | 1 | test | 0.269 | 0.302 | 0.192 | 3 | 2 | test | 0.269 | 0.302 | 0.192 |
| 5 | 1 | test | 0.272 | 0.305 | 0.194 | 5 | 2 | test | 0.272 | 0.305 | 0.194 |

**Table 4.** Performance comparison on the COCOPersons dataset [24] over the number of iterations $N_{TPR}$ and $N_{GBSR}$.

| $N_{TPR}$ | $N_{GBSR}$ | $AP$ | $AP_M$ | $AP_L$ | $N_{TPA}$ | $N_{GBSA}$ | $AP$ | $AP_M$ | $AP_L$ |
|-----------|------------|------|--------|--------|-----------|------------|------|--------|--------|
| 1 | 1 | 0.598 | 0.539 | 0.696 | 1 | 2 | 0.598 | 0.539 | 0.696 |
| 3 | 1 | 0.618 | 0.558 | 0.709 | 3 | 2 | 0.618 | 0.558 | 0.709 |
| 5 | 1 | 0.626 | 0.565 | 0.714 | 5 | 2 | 0.626 | 0.565 | 0.714 |

rounds in TPR Module demonstrates that the optical flow estimations are also improved during the mutual refinement process.

**4.4.2   The improvements brought by TPR Module and GBSR Module** The framework is composed of TPR Module and GBSR Module. To demonstrate the merits of both modules, we compare the performance of using neither of them (only using Seg Module and Pose Module), using TPR Module together with Pose Module and using TPR Module together with GBSR Module. According to the discussion section in 4.4.1, $N_{TPR} = 5$ and $N_{GBSA} = 1$. Table 5 shows the results on the OCHuman dataset while Table 6 shows the results on the COCOPersons dataset. Using Seg Module or Pose Module means only using the Seg Module in Fig. 5 or the Pose Module in Fig. 3 without other components and do not introduce iterations. It can be seen that both TPR Module and GBSR Module contribute to improvements on performance.

**Table 5.** Influence on the validation / test set of the OCHuman dataset [35] brought by TPR Module and GBSR Module.

| Configuration | Split | $AP$ | $AP_M$ | $AP_H$ |
|---|---|---|---|---|
| Seg Module and Pose Module | val | 0.223 | 0.262 | 0.150 |
| TPR Module and Pose Module | val | 0.237 | 0.277 | 0.163 |
| TPR Module and GBSR Module | val | 0.267 | 0.310 | 0.181 |
| Seg Module and Pose Module | test | 0.239 | 0.268 | 0.175 |
| TPR Module and Pose Module | test | 0.253 | 0.285 | 0.186 |
| TPR Module and GBSR Module | test | 0.272 | 0.305 | 0.194 |

**Table 6.** Influence on the COCOPersons dataset [24] brought by TPR Module and GBSR Module.

| Configuration | $AP$ | $AP_M$ | $AP_H$ |
|---|---|---|---|
| Seg Module and Pose Module | 0.555 | 0.498 | 0.670 |
| TPR Module and Pose Module | 0.576 | 0.519 | 0.682 |
| TPR Module and GBSR Module | 0.626 | 0.565 | 0.714 |

The complementary nature of pose estimation and segmentation has already been demonstrated in [35] and it is necessary to integrate GBSR Module in the framework.

**4.4.3  The influence of temporal window size** In Fig. 3 and Fig. 5, three consecutive images are fed into TPR and GBSR once at a time. Experiments are conducted to evaluate the influence of temporal window size on performance. $N_{TPA} = 5$ and $N_{GBSA} = 1$ are fixed. If the window size is 2, the $AP$ on the validation set and test set of OCHuman dataset [35] drop to 0.264 and 0.268, respectively. The $AP$ on the COCOPersons dataset [24] drops to 0.612. Due to limitations in computational resources, the temporal window size is not further enlarged.

## 5   Conclusions

The paper presents a novel framework for instance segmentation. The proposed TPR Module conducts mutual refinement between segmentation and optical flow estimation while the GBSR Module refines pose estimations by enforcing the consistency among human graph structures from consecutive frames. Different coherent modules are unified in a framework and produce the final segmentation results. Experimental results on the OCHuman dataset [35] and the COCOPersons dataset [24] have shown that the proposed framework outperforms existing methods on the task by leveraging unlabelled data together with prior knowledge. The self-supervised learning process benefits from the prior knowledge about video data.

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
