# OpenReview forum: "A Self-Supervised Framework for Human Instance Segmentation"
_thecvf.com/ECCV/2020/Workshop/VIPriors — VIPriors Poster_

### Official Review · AnonReviewer1 · 2020-07-22
**Good implementation of temporal consistency, but motivation needs work**

**Confidence:** 4
**Rating:** 6

**Review:**

[Summary] In 2-3 sentences, describe the key ideas, experiments, and their significance.

The authors use temporal consistency through pose estimation and optical flow estimation to improve human instance segmentation. Though this approach technically does not qualify as self-supervision, the temporal inductive priors are strong and show clear improvements over baselines.

[Strengths] What are the strengths of the paper? Clearly explain why these aspects of the paper are valuable.

- Simple, powerful idea to use temporal consistency prior.
- Decent implementations of each contribution, with proven results.

[Weaknesses] What are the weaknesses of the paper? Clearly explain why these aspects of the paper are weak.

- This method is not a self-supervision method. Self-supervision is when a DNN learns from mistakes in predictions on the input data. This method instead uses prior knowledge to fill in missing information within the model.
- Related to the previous point: the motivation is poorly formulated. It relies on a unfounded claim that self-supervision is superior to using prior knowledge, while the paper actually uses prior knowledge, specifically the temporal consistency prior. As a result the introduction uses vague statements without supporting evidence.

[Overall rating] Paper rating: Weak accept

[Detailed comments] Additional comments regarding the paper (e.g. typos or other possible improvements you would like to see for the camera-ready version of the paper, if any.)

- N_{GBSR} has zero influence in table 3 & 4. Why do the authors not discuss this? Why do they show it in tables 3 & 4?
- Unsubstantiated claims: lines 73
- Fourth contribution is not a contribution. The experiments serve to prove your first contribution.
- Line 398: not clear why the different modules get "different data".
- Grammar: lines 39 "have",
- Typos: lines 87 "vetors"
- Line 486: "subjective" = not a fact != about subjects
- Start new sentences instead of using comma: lines 43, 70, 312, 403,
- Premature end to sentence: line 396
- Capitalization: line 398, 434

---

### Official Review · AnonReviewer2 · 2020-07-27
**A Self-Supervised Framework for Human Instance Segmentation**

**Confidence:** 4
**Rating:** 6

**Review:**

#### 1. [Summary] In 2-3 sentences, describe the key ideas, experiments, and their significance.
The paper proposes  human instance segmentation method which has two main modules: (i) mutual refinement between optical flow and segmentation, (ii) skeleton refinement module.

#### 2. [Strengths] What are the strengths of the paper? Clearly explain why these aspects of the paper are valuable.
- Refinement modules
- Performance
- Robustness to occlusion

#### 3. [Weaknesses] What are the weaknesses of the paper? Clearly explain why these aspects of the paper are weak.
- It is emphasized that the method does self supervised learning, but it is not clear how and where it happens.
- There is no clear information about online collected videos, their labeling process, how they are used during training and the contribution of this dataset.

#### 4. [Overall rating] Paper rating
6

#### 5. [Justification of rating] Please explain how the strengths and weaknesses aforementioned were weighed in for the rating.
A nicely engineered paper which uses some prior knowledge and refinements to improve the performance.

#### 6. [Detailed comments] Additional comments regarding the paper (e.g. typos or other possible improvements you would like to see for the camera-ready version of the paper, if any.)
- L.82: If the priors can only function as weak constraints how do they help in this paper?
- L.124: GBSR, TPR and Pose2Seg are not mentioned before and in this line, abbreviation is given. It can be better to give full form.
- Fig.3: Loss term is not visible.
- L.269: "..if without occlusion."
- L.455: How exactly are the annotations generated?
- L.462: What are those benchmark datasets?
- No explanation of Table 2 in the text.
- What are the limitations?
- What are the failing cases?
- How are memory and time consumption?

Typos:
- L.23: Unlabeled
- L.41: instance
- L.87: vectors
- L.279: subscripts of 'l' are different than on the Eq.2.
- I guess there is no reason to write methods and backbones twice in Table 1.

Citations:
- L.88: Wasserstein metrics

---

### Decision · Program_Chairs · 2020-07-29

**Decision:**

Accept (Poster)

**Comment:**

It is our pleasure to inform you that your paper has been accepted to the poster track of 1st Visual Inductive Priors for Data-Efficient Deep Learning Workshop.

Please note the following deadlines:
* August 11, 2020 - workshop material, including:
 * paper in PDF format;
 * pre-recorded video presentation;
 * slides of the presentation in PDF.
* September 15, 2020 - camera-ready paper

The reviews can be found on OpenReview. Please take these comments and suggestions into account when preparing the camera-ready version of your paper, which is due September 15, 2020. The camera-ready paper should be uploaded to OpenReview.

As part of the workshop, each accepted paper must submit a pre-recorded 90 second talk before August 11, 2020. You will receive more information on how to upload the material shortly. The requirements for the video are:
* Duration: maximum 90 seconds
* MP4 format
* File size max. 100 MB
* Has an inset with a video of the speaker
* 16:9 aspect ratio (strongly preferred)
* 1920x1080 resolution (strongly preferred, at least 720 height)

Our suggested software for pre-recording your presentation is Zoom. For more information, please refer to the following guides:
How to record with Zoom Guide: http://homepages.inf.ed.ac.uk/rbf/ECCV2020HowtoRecordusingZoom.pdf
How to Record with Zoom tutorial: https://www.youtube.com/watch?v=CR199W7HdC0
Please ensure that at least one of the authors of the paper is available to attend the workshop during the allotted times. Note that the workshop will take place in two sessions spread across time zones (details are to follow). We will send instructions on how to connect to the workshop as soon as possible. The schedule for all talks and papers will be posted soon at the workshop website: https://vipriors.github.io.

We look forward to seeing you at the workshop!